# Partitioned Learned Bloom Filter

**Kapil Vaidya**
CSAIL
MIT
kapilv@mit.edu

**Eric Knorr**
School of Engineering and Applied Sciences
Harvard University
eric.r.knorr@gmail.com

**Tim Kraska**
CSAIL
MIT
kraska@mit.edu

**Michael Mitzenmacher**
School of Engineering and Applied Sciences
Harvard University
michaelm@eecs.harvard.edu

## Abstract

Bloom filters are space-efficient probabilistic data structures that are used to test whether an element is a member of a set, and may return false positives. Recently, variations referred to as learned Bloom filters were developed that can provide improved performance in terms of the rate of false positives, by using a learned model for the represented set. However, previous methods for learned Bloom filters do not take full advantage of the learned model. Here we show how to frame the problem of optimal model utilization as an optimization problem, and using our framework derive algorithms that can achieve near-optimal performance in many cases. Experimental results from both simulated and real-world datasets show significant performance improvements from our optimization approach over both the original learned Bloom filter constructions and previously proposed heuristic improvements.

## 1 Introduction

Bloom filters are space-efficient probabilistic data structures that are used to test whether an element is a member of a set [Bloom (1970)]. A Bloom filter compresses a given set $S$ into an array of bits. A Bloom filter may allow false positives, but will not give false negative matches, which makes them suitable for numerous memory-constrained applications in networks, databases, and other systems areas. Indeed, there are many thousands of papers describing applications of Bloom filters [Dayan et al. (2018), Dillinger & Manolios (2004), Broder & Mitzenmacher (2003)].

There exists a trade off between the false positive rate and the size of a Bloom filter (smaller false positive rate leads to larger Bloom filters). For a given false positive rate, there are known theoretical lower bounds on the space used [Pagh et al. (2005)] by the Bloom filter. However, these lower bounds assume the Bloom filter could store any possible set. If the data set or the membership queries have specific structure, it may be possible to beat the lower bounds in practice [Mitzenmacher (2002), Bruck et al. (2006), Mitzenmacher et al. (2020)]. In particular, [Kraska et al. (2018)] and [Mitzenmacher (2018)] propose using machine learning models to reduce the space further, by using a learned model to provide a suitable pre-filter for the membership queries. This allows one to beat the space lower bounds by leveraging the context specific information present in the learned model. Rae et al. (2019) propose a neural Bloom Filter that learns to write to memory using a distributed write scheme and achieves compression gains over the classical Bloom filter.

The key idea of learned Bloom filters is that in many practical settings, given a query input, the likelihood that the input is in the set $S$ can be deduced by some observable features which can be captured by a machine learning model. For example, a Bloom filter that represents a set of malicious URLs can benefit from a learned model that can distinguish malicious URLs from benign URLs. This model can be trained on URL features such as length of hostname, counts of special characters, etc. This approach is described in [Kraska et al. (2018)], which studies how standard index structures can be improved using machine learning models; we refer to their framework as the original learned

Bloom filter, Given an input $x$ and its features, the model outputs a score $s(x)$ which is supposed to correlate with the likelihood of the input being in the set. Thus, the elements of the set, or keys, should have a higher score value compared to non-keys. This model is used as a pre-filter, so when score $s(x)$ of an input $x$ is above a pre-determined threshold $t$, it is directly classified as being in the set. For inputs where $s(x) < t$, a smaller backup Bloom filter built from only keys with a score below the threshold (which are known) is used. This maintains the property that there are no false negatives. The design essentially uses the model to immediately answer for inputs with high score whereas the rest of the inputs are handled by the backup Bloom filter as shown in Fig.1(A). The threshold value $t$ is used to partition the space of scores into two regions, with inputs being processed differently depending on in which region its score falls. With a sufficiently accurate model, the size of the backup Bloom filter can be reduced significantly over the size of a standard Bloom filter while maintaining overall accuracy. [Kraska et al. (2018)] showed that, in some applications, even after taking the size of the model into account, the learned Bloom filter can be smaller than the standard Bloom filter for the same false positive rate.

The original learned Bloom filter compares the model score against a single threshold, but the framework has several drawbacks.

**Choosing the right threshold**: The choice of threshold value for the learned Bloom filter is critical, but the original design uses heuristics to determine the threshold value.

**Using more partitions:** Comparing the score value only against a single threshold value wastes information provided by the learning model. For instance, two elements $x_1, x_2$ with $s(x_1) \gg s(x_2) > t$, are treated the same way but the odds of $x_1$ being a key are much higher than for $x_2$. Intuitively, we should be able to do better by partitioning the score space into more than two regions.

**Optimal Bloom filters for each region:** Elements with scores above the threshold are directly accepted as keys. A more general design would provide backup Bloom filters in both regions and choose the Bloom filter false positive rate of each region so as to optimize the space/false positive trade-off as desired. The original setup can be interpreted as using a Bloom filter of size 0 and false positive rate of 1 above the threshold. This may not be the optimal choice; moreover, as we show, using different Bloom filters for each region(as shown in Fig.1(C)) allows further gains when we increase the number of partitions.

Follow-up work by [Mitzenmacher (2018)] and [Dai & Shrivastava (2019)] improve on the original design but only address a subset of these drawbacks. In particular, [Mitzenmacher (2018)] proposes using Bloom filters for both regions and provides a method to find the optimal false positive rates for each Bloom filter. But [Mitzenmacher (2018)] only considers two regions and does not consider how to find the optimal threshold value. [Dai & Shrivastava (2019)] propose using multiple thresholds to divide the space of scores into multiple regions, with a different backup Bloom filter for each score region. The false positive rates for each of the backup Bloom filters and the threshold values are chosen using heuristics. Empirically, we found that these heuristics might perform worse than [Mitzenmacher (2018)] in some scenarios.

A general design that resolves all the drawbacks would, given a target false positive rate and the learned model, partition the score space into multiple regions with separate backup Bloom filters for each region, and find the optimal threshold values and false positive rates, under the goal of minimizing the memory usage while achieving the desired false positive rate as shown in Fig.1(C). In this work, we show how to frame this problem as an optimization problem, and show that our resulting solution significantly outperforms the heuristics used in previous works. Additionally, we show that our maximum space saving[1] is linearly proportional to the KL divergence of the key and non-key score distributions determined by the partitions. We present a dynamic programming algorithm to find the optimal parameters (up to the discretization used for the dynamic programming) and demonstrate performance improvements over a synthetic dataset and two real world datasets: URLs and EMBER. We also show that the performance of the learned Bloom filter improves with increasing number of partitions and that in practice a small number of regions ($\approx 4 - 6$) suffices to get a very good performance. We refer to our approach as a *partitioned learned Bloom filter* (PLBF). Experimental results from both simulated and real-world datasets show significant performance improvements. We show that to achieve a false positive rate of $0.001$, [Mitzenmacher (2018)] uses

---

[1]space saved by using our approach instead of a Bloom filter

8.8x, 3.3x and 1.2x the amount of space and [Dai & Shrivastava (2019)] uses 6x, 2.5x and 1.1x the amount of space compared to PLBF for synthetic, URLs and EMBER respectively.

## 2 BACKGROUND

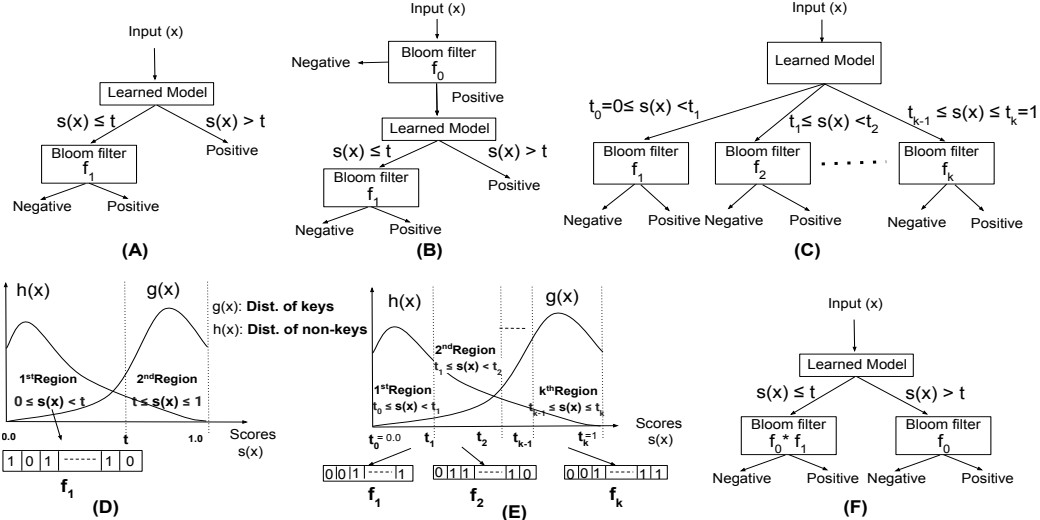

Figure 1: (A),(B),(C) represent the original LBF, LBF with sandwiching, and PLBF designs, respectively. Each region in (C) is defined by score boundaries $t_i, t_{i+1}$ and a false positive rate $f_i$ of the Bloom Filter used for that region. (D),(E) show the LBF and PLBF with score space distributions. (F) represents a PLBF design equivalent to the sandwiching approach used in Appendix.D.1.

### 2.1 STANDARD BLOOM FILTERS AND RELATED VARIANTS

A standard Bloom filter, as described in Bloom's original paper [Bloom (1970)], is for a set $S = \{x_1, x_2, ..., x_n\}$ of $n$ keys. It consists of an array of $m$ bits and uses $k$ independent hash functions $\{h_1, h_2, ...h_k\}$ with the range of each $h_i$ being integer values between 0 and $m - 1$. We assume the hash functions are fully random. Initially all $m$ bits are 0. For every key $x \in S$, array bits $h_i(x)$ are set to 1 for all $i \in \{1, 2, ...k\}$.

A membership query for $y$ returns that $y \in S$ if $h_i(y) = 1$ for all $i \in \{1, 2, ...k\}$ and $y \notin S$ otherwise. This ensures that the Bloom filter has no false negatives but non-keys $y$ might result in a false positive. This false positive rate depends on the space $m$ used by the Bloom Filter. Asymptotically (for large $m, n$ with $m/n$ held constant), the false positive rate is given by

$$\left(1 - \left(1 - \frac{1}{m}\right)^{kn}\right)^k. \tag{1}$$

See [Broder & Mitzenmacher (2003); Bose et al. (2008)] for further details.

[Bloom (1970)] proved a space lower bound of $|S| \times \log_2(\frac{1}{F})$ for a Bloom filter with false positive rate $F$. The standard construction uses space that is asymptotically $\log_2 e(\approx 1.44)$ times more than the lower bound. Other constructions exist, such as Cuckoo filters[Fan et al. (2014)], Morton filters[Breslow & Jayasena (2018)], XOR filters[Graf & Lemire (2020)] and Vacuum filters[Wang et al. (2019)]. These variants achieve slightly better space performance compared to standard Bloom filters but still are a constant factor larger than the lower bound. [Pagh et al. (2005)] presents a Bloom filter design that achieves this space lower bound, but it appears too complicated to use in practice.

### 2.2 LEARNED BLOOM FILTER

Learned Bloom filters make use of learned models to beat the theoretical space bounds. Given a learned model that can distinguish between keys and non-keys, learned Bloom filters use it as a pre-filter before using backup Bloom filters. The backup Bloom filters can be any variant including the standard, cuckoo, XOR filters, etc. If the size of the model is sufficiently small, learned models can be used to enhance the performance of any Bloom filter variant.

We provide the framework for learned Bloom filters. We are given a set of keys $S = \{x_1, x_2, .., x_n\}$ from a universe $U$ for which to build a Bloom filter. We are also given a sample of the non-keys $Q$ which is representative of the set $U - S$. Features that can help in determining if an element is a member of $S$ are determined. The learned model is then trained on features of set $S \cup Q$ for a binary classification task and produces a score $s(x) \in [0, 1]$. This score $s(x)$ can be viewed (intuitively, not formally) as the confidence of the model that the element $x$ is in the set $S$. So, a key in $S$ would ideally have a higher score value than the non-keys. An assumption in this framework is that the training sample distribution needs to match or be close to the test distribution of non-keys; the importance of this assumptions has been discussed at length in [Mitzenmacher (2018)]. For many applications, past workloads or historical data can be used to get an appropriate non-key sample.

As discussed above, [Kraska et al. (2018)] set a threshold $t$ and inputs satisfying $s(x) > t$ are classified as a key. A backup Bloom filter is built for just the keys in $S$ satisfying $s(x) \le t$. This design is represented in Fig.1(A). [Mitzenmacher (2018)] proposes using another Bloom filter *before* the learned model along with a backup Bloom Filter. As the learned model is used between two Bloom filters as shown in Fig.1(B), this is referred to as the 'sandwiching' approach. They also provide the analysis of the optimal false positive rates for a given amount of memory for the two Bloom filters (given the false negative rate and false positive rate for the learned model, and the corresponding threshold). Interestingly, the sandwiching approach and analysis can be seen as a special case of our approach and analysis, as we describe later in Appendix.D.1. [Dai & Shrivastava (2019)] use multiple thresholds to partition the score space into multiple regions and use a backup Bloom filter for each score region. They propose heuristics for how to divide up the score range and choose false positive rate per region.

## 3 PARTITIONED LEARNED BLOOM FILTER (PLBF)

### 3.1 DESIGN

As discussed before, the general design segments the score space into multiple regions using multiple thresholds, as shown in Fig.1(C), and uses separate backup Bloom filters for each region. We can choose different target false positive rates for each region[2]. The parameters associated with each region are its threshold boundaries and its false positive rate. Setting good values for these parameters is crucial for performance. Our aim is to analyze the performance of the learned Bloom filter with respect to these parameters, and find methods to determine optimal or near-optimal parameters.

The following notation will be important for our analysis. Let $G(t)$ be the fraction of keys with scores falling below $t$. We note that since the key set is finite, $G(t)$ goes through discrete jumps. But it is helpful (particularly in our pictures) to think of $G(t)$ as being a continuous function, corresponding to a cumulative probability distribution, with a corresponding "density" function $g(t)$. For non keys, we assume that queries involving non-keys come from some distribution $\mathcal{D}$, and we define $H(t)$ to be probability that a non-key query from $\mathcal{D}$ has a score less than or equal to $t$. Note that non key query distribution might be different from non key distribution. If non key queries are chosen uniformly at random, non key query distribution would be the same as non key distribution. We assume that $H(t)$ is known in the theoretical analysis below. In practice, we expect a good approximation of $H(t)$ will be used, determined by taking samples from $\mathcal{D}$ or a suitably good approximation, which may be based on, for example, historical data (discussed in detail in [Mitzenmacher (2018)]). Here $H(t)$ can be viewed as a cumulative distribution function, and again in our pictures we think of it as having a density $h(t)$. Also, note that if queries for non-keys are simply chosen uniformly at random, then $H(t)$ is just the fraction of non-keys with scores below $t$. While our analysis holds generally, the example of $H(t)$ being the fraction of non-keys with scores below $t$ may be easier to keep in mind. Visualization of the original learned Bloom filter in terms of these distributions is shown in Fig.1(D).

As we describe further below, for our partitioned learned Bloom filter, we use multiple thresholds and a separate backup Bloom filter for each region, as show in Fig.1($E$). In what follows, we formulate the problem of choosing thresholds and backup Bloom filter false positive rates (or equivalently, sizes) as an optimization problem in section 3.2. In section 3.3.1, we find the optimal solution of a relaxed problem which helps us gain some insight into the general problem. We then propose an approximate solution for the general problem in section 3.3.3.

---

[2]The different false positive rates per region can be achieved in multiple ways. Either by choosing a separate Bloom filter per region or by having a common Bloom filter with varying number of hash functions per region.

We find in our formulation that the resulting parameters correspond to quite natural quantities in terms of $G$ and $H$. Specifically, the optimal false positive rate of a region is proportional to the ratio of the fraction of keys to the fraction of non-keys in that region. If we think of these region-based fractions for keys and non-keys as probability distributions, the maximum space saving obtained is proportional to the KL divergence between these distributions. Hence we can optimize the thresholds by choosing them to maximize this divergence. We show that we can find thresholds to maximize this divergence, approximately, through dynamic programming. We also show that, naturally, this KL divergence increases with more number of regions and so does the performance. In our experiments, we find a small number($\approx 4 - 6$) of partitions suffices to get good performance.

## 3.2 GENERAL OPTIMIZATION FORMULATION

To formulate the overall problem as an optimization problem, we consider the variant which minimizes the space used by the Bloom filters in PLBF in order to achieve an overall a target false positive rate ($F$). We could have similarly framed it as minimizing the false positive rate given a fixed amount of space. Here we are assuming the learned model is given.

We assume normalized score values in $[0, 1]$ for convenience. We have region boundaries given by $t_i$ values $0 = t_0 \le t_1 \le \dots t_{k-1} \le t_k = 1$, with score values between $[t_{i-1}, t_i]$ falling into the $i^{th}$ region. We assume the target number of regions $k$ is given. We denote the false positive rate for the Bloom filter in the $i^{th}$ region by $f_i$. We let $G$ and $H$ be defined as above. As state previously, Fig.1($E$) corresponds to this setting, and the following optimization problem finds the optimal thresholds $t_i$ and the false positive rates $f_i$:

$$\min_{t_i, f_i} \quad \left( \sum_{i=1}^{k} |S| \times (G(t_i) - G(t_{i-1})) \times c \log_2 \left( \frac{1}{f_i} \right) \right) \text{ + Size of Learned Model} \qquad (2)$$

$$\text{constraints} \quad \sum_{i=1}^{k} (H(t_i) - H(t_{i-1})) \times f_i \le F \qquad (3)$$

$$f_i \le 1 \ , i = 1 \dots k \qquad (4)$$

$$(t_i - t_{i-1}) \ge 0 \ , i = 1 \dots k \ ; \ t_0 = 0; t_k = 1 \qquad (5)$$

The minimized term (Eq.2) represents the total size of the learned Bloom filter, the size of backup Bloom filters is obtained by summing the individual backup Bloom filter sizes. The constant $c$ in the equation depends on which variant of the Bloom filter is used as the backup[3]; as it happens, its value will not affect the optimization.

The first constraint (Eq.3) ensures that the overall false positive rate stays below the target $F$. The overall false positive rate is obtained by summing the appropriately weighted rates of each region. The next constraint (Eq.4) encodes the constraint that false positive rate for each region is at most 1. The last set of constraints (Eq.5) ensure threshold values are increasing and cover the interval $[0, 1]$.

## 3.3 SOLVING THE OPTIMIZATION PROBLEM

### 3.3.1 SOLVING A RELAXED PROBLEM

If we remove the false positive rate constraints (Eq.4, giving $f_i \le 1$), we obtain a relaxed problem shown in Eq.6. This relaxation is useful because it allows us to use the Karush-Kuhn-Tucker (KKT) conditions to obtain optimal $f_i$ values in terms of the $t_i$ values, which we used to design algorithms for finding near-optimal solutions. Throughout this section, we assume the the relaxed problem yields a solution for the original problem; we return to this issue in subsection 3.3.3.

$$\min_{t_{i=1 \dots k-1}, f_{i=1 \dots k}} \quad \left( \sum_{i=1}^{k} |S| \times (G(t_i) - G(t_{i-1})) \times c \log_2 \left( \frac{1}{f_i} \right) \right) \text{ + Size of Learned Model}$$

$$\text{constraints} \quad \sum_{i=1}^{k} (H(t_i) - H(t_{i-1})) \times f_i \le F;$$

$$(t_i - t_{i-1}) \ge 0 \ , i = 1 \dots k; \ t_0 = 0; \ t_k = 1$$

$$(6)$$

---

[3]The sizes of Bloom filter variants are proportional to $|S| \times \log_2(1/f)$, where $S$ is the set it represents, and $f$ is the false positive rate it achieves. See e.g. [Mitzenmacher (2018)] for related discussion. The constant $c$ depends on which type of Bloom filter is used as a backup. For example, $c = \log_2(e)$ for standard Bloom filter.

The optimal $f_i$ values obtained by using the KKT conditions yield Eq.7 (as derived in Appendix.A), giving the exact solution in terms of $t_i$'s.

$$f_i = F \frac{G(t_i) - G(t_{i-1})}{H(t_i) - H(t_{i-1})} \tag{7}$$

The numerator $G(t_i) - G(t_{i-1})$ is the fraction of keys in the $i^{th}$ region and the denominator $H(t_i) - H(t_{i-1})$ is the probability of a non-key query being in the $i^{th}$ region. In intuitive terms, the false positive rate for a region is proportional to the ratio of the key density (fraction of keys) to non-key density (fraction of non-key queries). Since we have found the optimal $f_i$ in terms of the $t_i$, we can replace the $f_i$ in the original problem to obtain a problem only in terms of the $t_i$. In what follows, we use $g(\hat{\mathbf{t}})$ to represent the discrete distribution given by the $k$ values of $G(t_i) - G(t_{i-1})$ for $i = 1, \ldots, k$, and similarly we use $h(\hat{\mathbf{t}})$ for the distribution corresponding to the $H(t_i) - H(t_{i-1})$ values. Eq.8 shows the rearrangement of the minimization term(excluding model size) after substitution.

$$\begin{aligned}
\text{Min. Term} &= \sum_{i=1}^{k} |S| \times (G(t_i) - G(t_{i-1})) \times c \log_2 \left( \frac{H(t_i) - H(t_{i-1})}{(G(t_i) - G(t_{i-1})) \times F} \right) \\
&= \sum_{i=1}^{k} |S| \times (G(t_i) - G(t_{i-1})) \times c \log_2 \left( \frac{1}{F} \right) - c \times |S| \times D_{KL} \left( g(\hat{\mathbf{t}}), h(\hat{\mathbf{t}}) \right)
\end{aligned} \tag{8}$$

where $D_{KL}$ is the standard KL divergence for the distributions given by $g(\hat{\mathbf{t}})$ and $h(\hat{\mathbf{t}})$.

Eq.8 represents the space occupied by the backup Bloom filters; the total space includes this and the space occupied by the learned model.

$$c \times \left( |S| \times \log_2 \left( \frac{1}{F} \right) - |S| \times D_{KL} \left( g(\hat{\mathbf{t}}), h(\hat{\mathbf{t}}) \right) \right) + \text{Size Of Learned Model} \tag{9}$$

The space occupied by the Bloom filter without the learned model is $c \times |S| \times \log_2(1/F)$. Thus, the space saved by PLBF in comparison to the normal Bloom filter is:

$$c \times \left( |S| \times D_{KL} \left( g(\hat{\mathbf{t}}), h(\hat{\mathbf{t}}) \right) \right) - \text{Size Of Learned Model} \tag{10}$$

The space saved by PLBF is therefore linearly proportional to the KL divergence of key and non-key distributions of the regions given by $g(\hat{\mathbf{t}})$ and $h(\hat{\mathbf{t}})$ of the regions.

This derivation suggests that the KL divergence might also be used as a loss function to improve the model quality. We have tested this empirically, but thus far have not seen significant improvements over the MSE loss we use in our experiments; this remains an interesting issue for future work.

### 3.3.2 FINDING THE OPTIMAL THRESHOLDS FOR RELAXED PROBLEM

We have shown that, given a set of thresholds, we can find the optimal false positive rates for the relaxed problem. Here we turn to the question of finding optimal thresholds. We assume again that we are given $k$, the number of regions desired. (We consider the importance of choosing $k$ further in our experimental section.) Given our results above, the optimal thresholds correspond to the points that maximize the KL divergence between $(g(\hat{\mathbf{t}}), h(\hat{\mathbf{t}}))$. The KL divergence of $(g(\hat{\mathbf{t}}), h(\hat{\mathbf{t}}))$ is the sum of the terms $g_i \log_2 \frac{g_i}{h_i}$, one term per region. (Here $g_i = G(t_i) - G(t_{i-1})$ and $h_i = H(t_i) - H(t_{i-1})$.) Note that each term depends only on the proportion of keys and non-keys in that region and is otherwise independent of the other regions. This property allows a recursive definition of KL divergence that is suitable for dynamic programming.

We divide the score space $[0, 1]$ into $N$ consecutive small segments for a chosen value of $N$; this provides us a discretization of the score space, with larger $N$ more closely approximating the real interval. Given $k$, we can find a set of $k$ approximately optimal thresholds using dynamic programming, where the solution is approximate due to our discretization of the score space. Let $DP_{KL}(n, j)$ denote the maximum divergence one can get when you divide the first $n$ segments into $j$ regions. Our approximately optimal divergence corresponds to $DP_{KL}(N, k)$. The idea behind the algorithm is that the we can recursively define $DP_{KL}(n, j)$ as represented in Eq.11. Here $g', h'$ represent the fraction of keys and the fraction of non-key queries, respectively, in these $N$ segments.

$$DP_{KL}\left(n, j\right) = \max\left(DP_{KL}(n-i, j-1) + \left(\sum_{r=i}^{n} g'(r) \times \log_2\left(\frac{\sum_{r=i}^{n} g'(r)}{\sum_{r=i}^{n} h'(r)}\right)\right)\right) \quad (11)$$

The time complexity of computing $DP_{KL}(N, k)$ is $\mathcal{O}(N^2 k)$. One can increase the value of $N$ to get more precision in the discretization when finding thresholds, at the cost of higher computation time.

### 3.3.3 THE RELAXED PROBLEM AND THE GENERAL PROBLEM

We can find a near-optimal solution to the relaxed problem by first, obtaining the threshold values that maximize the divergence and then, getting the optimal $f_i$ values using Eq.7. In many cases, the optimal relaxed solution will also be the optimal general solution, specifically if $F \times (G(t_{i-1}) - G(t_i))/(H(t_{i-1}) - H(t_i)) < 1$ for all $i$. Hence, if we are aiming for a sufficiently low false positive rate $F$, solving the relaxed problem suffices.

To solve the general problem, we need to deal with regions where $f_i \geq 1$, but we can use the relaxed problem as a subroutine. First, given a fixed set of $t_i$ values for the general problem, we have an algorithm (Alg.1, as discussed in Appendix.B) to find the optimal $f_i$'s. Briefly summarized, we solve the relaxed problem, and for regions with $f_i > 1$, the algorithm sets $f_i = 1$, and then re-solves the relaxed problem with these additional constraints, and does this iteratively until no region with $f_i > 1$ remains. The problem is that we do not have the optimal set of $t_i$ values to begin; as such, we use the optimal $t_i$ values for the relaxed solution as described in Section 3.3.2. This yields a solution to the general problem (psuedo-code in Alg.2), but we emphasize that it is not optimal in general, since we did not start with the optimal $t_i$. We expect still that it will perform very well in most cases.

In practice, we observe that keys are more concentrated on higher scores, and non-key queries are more concentrated on lower scores. Given this property, if a region with $f_i = 1$ (no backup Bloom filter used) exists in the optimal solution of the general problem, it will most probably be the rightmost region. In particular, if $(G(t_{i-1}) - G(t_i))/(H(t_{i-1}) - H(t_i))$ is increasing as $t_{i-1}, t_i$ increase – that is, the ratio of the fraction of keys to the fraction of non-key queries over regions is increasing – then indeed without loss of generality the last ($k$th) region will be the only one with $f_k = 1$. (We say only one region because any two consecutive regions with $f_i = 1$ can be merged and an extra region can be made in the remaining space which is strictly better, as adding an extra region always helps as shown in Appendix.D.2.) It is reasonable to believe that in practice this ratio will be increasing or nearly so.

Hence if we make the assumption that in the optimal solution all the regions except the last satisfy the $f_i < 1$ constraint, then if we identify the optimal last region's boundary, we can remove the $f_i \leq 1$ constraints for $i \neq k$ and apply the DP algorithm to find near optimal $t_i$'s. To identify the optimal last region's boundary, we simply try all possible boundaries for the $k$th region (details discussed in Appendix.C). As it involves assumptions on the behavior of $G$ and $H$, we emphasize again that this will not guarantee finding the optimal solution. But when the conditions are met it will lead to a near-optimal solution (only near-optimal due to the discretization of the dynamic program).

## 4 EVALUATION

We compare PLBF against the theoretically optimal Bloom filter [Bloom (1970)][4], the sandwiching approach [Mitzenmacher (2018)], and AdaBF [Dai & Shrivastava (2019)]. Comparisons against standard Bloom filters[5] appear in Appendix.E.1. We excluded the original learned Bloom filter [Kraska et al. (2018)] as the sandwiching approach was strictly better. We include the size of the learned model with the size of the learned Bloom filter. To ensure a fair comparison, we used the optimal Bloom filter as the backup bloom filter for all learned variants. We use 3 different datasets:

**URLs**: As in previous papers [Kraska et al. (2018), Dai & Shrivastava (2019)], we used the URL data set, which contains 103520 (23%) malicious and 346646 (77%) are benign URLs. We used 17 features from these URL's such as host name length, use of shortening, counts of special characters,etc.

---

[4]For the space of a theoretically optimal Bloom filter, we take the standard Bloom filter of same false positive rate and divide it's space used by $\log_2 e$, as obtaining near-optimality in practice is difficult. This uses the fact that the standard Bloom filter is asymptotically $\log_2 e$ times suboptimal than the optimal as discussed in Sec.2.1.

[5]PLBF performs better against standard Bloom filters, as discussed in Appendix.D.3. Section 4.1 are conservative estimates of gains possible in practice using a PLBF.

[h!]

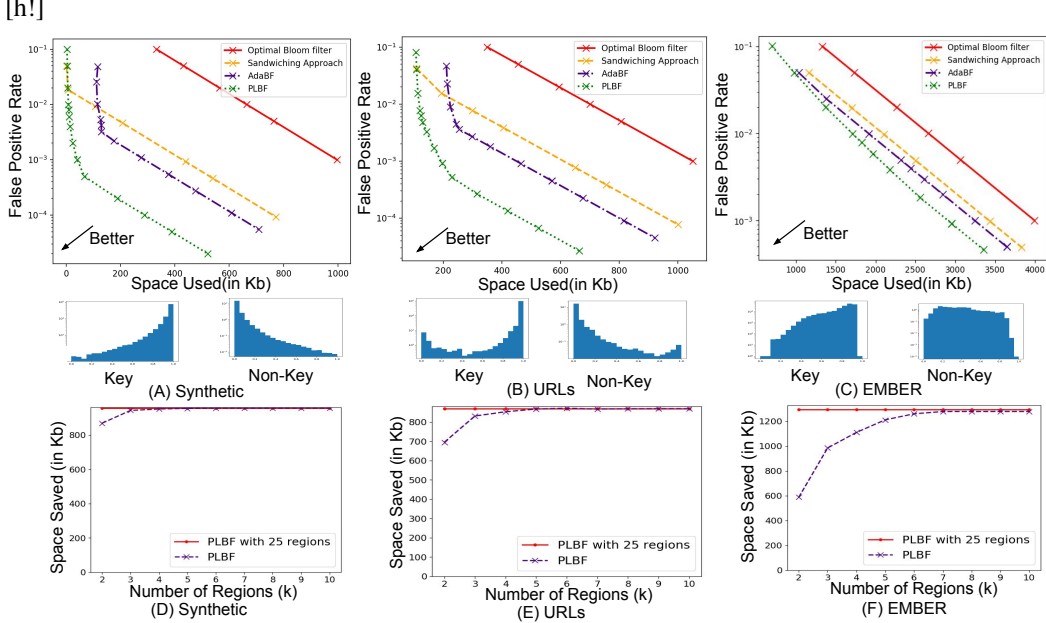

Figure 2: FPR vs Space for the (A) Synthetic (B) URLs (C) EMBER datasets for various baselines along with key and non-key score distributions. Space Saved as we increase number of regions for the (D) Synthetic (E) URLs (F) EMBER datasets for PLBF compared to the optimal Bloom filter

**EMBER**: Bloom filters are widely used to match file signatures with the virus signature database. Ember (Endgame Malware Benchmark for Research) [Anderson & Roth (2018)] is an open source collection of 1.1M sha256 file hashes that were scanned by VirusTotal in 2017. Out of the 1.1 million files, 400K are malicious, 400K are benign, and we ignore the remaining 300K unlabeled files. The features of the files are already included in the dataset.

**Synthetic**: An appealing scenario for our method is when the key density increases and non-key density decreases monotonically with respect to the score value. We simulate this by generating the key and non-key score distribution using Zipfian distributions as in Fig.2(A). Since we directly work on the score distribution, the size of the learned model for this synthetic dataset is zero.

## 4.1 OVERALL PERFORMANCE

Here, we compare the performance of PLBF against other baselines by fixing the target $F$ and measuring the space used by each methods. We use PLBF Alg.3 with DP algorithm discretization($N$) set to 1000. We train the model on the entire key set and 40% of the non-key set. The thresholds and backup Bloom filters are then tuned using this model with the aim of achieving the fixed *target F*. The rest of the non-keys are used to evaluate the *actual* false positive rate.

While any model can be used, we choose the random forest classifier from sklearn [Pedregosa et al.] for its good accuracy. The F1 scores of the learned models used for synthetic, URLs and EMBER were 0.99, 0.97, and 0.85, respectively. We consider the size of the model to be the pickle file size on the disk (a standard way of serializing objects in Python). We use five regions ($k = 5$) for both PLBF and AdaBF as this is usually enough to achieve good performance as discussed in 4.2. Using higher $k$ would only improve our performance.

The results of the experiment are shown in the Fig.2(A-C) along with the distribution of the scores of keys and non-keys for each dataset. As we can see from the figure, PLBF has a better Pareto curve than the other baselines for all the datasets. On the synthetic dataset and URLs dataset we observe a significantly better performance. In contrast, for the EMBER dataset our performance is only slightly better indicating that the model here is not as helpful. The difference between space used by PLBF and optimal Bloom filter first increases with decreasing false positive rate but converges to a constant value for all datasets, as given in Eq.10. For the same amount of space used(400Kb,500Kb,3000Kb space for synthetic,URLs,EMBER, respectively), PLBF achieves 22x, 26x, and 3x smaller false

positive rates than the sandwiching approach, and 8.5x, 9x, and 1.9x smaller false positive rates than AdaBF for synthetic, URLs, and EMBER, respectively. To achieve a false positive rate of $0.001$, the sandwiching approach uses 8.8x, 3.3x, and 1.2x the amount of space and AdaBF uses 6x, 2.5x, and 1.1x the amount of space compared to PLBF for synthetic, URLs, and EMBER datasets respectively.

## 4.2 PERFORMANCE AND THE NUMBER OF REGIONS

The maximum space savings obtained by using PLBF is linearly proportional to the KL divergence of the distributions(Eq10) and this KL divergence strictly increases with the number of regions(Appendix.D.2). Fig.2(D-F) show the space saved w.r.t the optimal Bloom filter as we increase the number of regions $k$ for a target false positive rate of $0.001$. The red line in the figure shows the savings when using 25 regions; using more regions provides no noticeable improvement on this data. Our results suggest using 4-6 regions should be sufficient to obtain reasonable performance. We have additional experiments in Appendix.E that shows PLBF performance against standard Bloom filters and PLBF performance w.r.t model quality.

## 5 CONCLUSION

Our analysis of the partitioned learned Bloom filter provides a formal framework for improving on learned Bloom filter performance that provides substantially better performance than previous heuristics. As Bloom filters are used across thousands of applications, we hope the PLBF may find many uses where the data set is amenable to a learned model.

Acknowledgments

This research was supported in part by NSF grants CCF-1563710 and CCF-1535795, and by a gift to the Center for Research on Computation and Society at Harvard University.

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

## A  SOLVING THE RELAXED PROBLEM USING KKT CONDITIONS

As mentioned in the main text, if we relax the constraint of $f_i \leq 1$, using the stationary KKT conditions we can obtain the optimal $f_i$ values. Here we show this work. The appropriate Lagrangian equation is given in Eq.12. In this case, the KKT coniditions tell us that the optimal solution is a stationary point of the Lagrangian. Therefore, we find where the derivative of the Lagrangian with respect to $f_i$ is zero.

$$L\left(t_i, f_i, \lambda, \nu_i\right) = \sum_{i=1}^{k} \left(G(t_i) - G(t_{i-1})\right) \times c \log_2\left(\frac{1}{f_i}\right) + \lambda \times \left(\left(\sum_{i=1}^{k}\left(H(t_i) - H(t_{i-1})\right) \times f_i\right) - F\right) + \sum_{i=1}^{k} \nu_i \times (t_{i-1} - t_i) \tag{12}$$

$$\frac{\partial L(t_i, f_i, \lambda, \nu_i)}{\partial f_i} = 0 \tag{13}$$

$$\frac{\partial (G(t_i) - G(t_{i-1})) c \log_2\left(\frac{1}{f_i}\right)}{\partial f_i} = -\lambda \frac{\partial (H(t_i) - H(t_{i-1})) \times f_i}{\partial f_i} \tag{14}$$

$$f_i = \frac{c \ln(2) \times (G(t_i) - G(t_{i-1})) \times \lambda}{(H(t_i) - H(t_{i-1}))} \tag{15}$$

$$\lambda = \frac{F}{c \ln(2) \times \sum_{i=1}^{k} (G(t_i) - G(t_{i-1}))} = \frac{F}{c \ln 2} \tag{16}$$

$$f_i = \frac{(G(t_i) - G(t_{i-1})) \times FPR}{(H(t_i) - H(t_{i-1}))} \tag{17}$$

Eq.15 expresses $fpr_i$ in terms of $\lambda$. Summing Eq.15 over all $i$ and using the relationship between $F$ and $H$ we get Eq.16. Thus the optimal $f_i$ values turn out to be as given in Eq.17.

---

**Algorithm 1** Finding optimal fpr's given thresholds

---

    **Input** $G'$ - the array containing key density of each region
    **Input** $H'$ - the array containing non-key density of each region
    **Input** $F$ - target overall false positive rate
    **Input** $k$ - number of regions
    **Output** $f$ - the array of false positive rate of each region

1: **procedure** OPTIMALFPR($G', H', F, k$)
2:     $G_{sum} \leftarrow 0$                                               ▷ sum of key density of regions with $f_i = 1$
3:     $H_{sum} \leftarrow 0$                                             ▷ sum of non-key density of regions with $f_i = 1$
4:     **for** $i$ **in** $1, 2, ...k$ **do**
5:         $f[i] \leftarrow \frac{G'[i] \cdot F}{H'[i]}$                                ▷ Assign relaxed problem solution
6:     **while** some $f[i] > 1$ **do**
7:         **for** $i$ **in** $1, 2, ...k$ **do**
8:             **if** $(f[i] > 1)$ **then** $f[i] \leftarrow 1$                ▷ Cap the false positive rate of region to one
9:         $G_{sum} \leftarrow 0$
10:        $H_{sum} \leftarrow 0$
11:        **for** $i$ **in** $1, 2, ...k$ **do**
12:            **if** $(f[i] = 1)$ **then** $G_{sum} \leftarrow G_{sum} + G'[i]; H_{sum} \leftarrow H_{sum} + H'[i]$     ▷ Calculate key,non-key density in regions
    with no Bloom filter($f[i] = 1$)
13:        **for** $i$ **in** $1, 2, ...k$ **do**
14:            **if** $(f[i] < 1)$ **then** $f[i] = \frac{G'[i] \cdot (F - H_{sum})}{H'[i] \cdot (1 - G_{sum})}$   ▷ Modifying the $f_i$ of the regions to ensure target false positive rate is FPR
15:     **return** $fpr$ Array

---

**Algorithm 2** Using relaxed solution for the general problem

---

    **Input** $G_{dis}$ - the array containing discretized key density of each region
    **Input** $H_{dis}$ - the array containing discretized key density of each region
    **Input** $F$ - target overall false positive rate
    **Input** $k$ - number of regions
    **Output** $t$ - the array of threshold boundaries of each region
    **Output** $f$ - the array of false positive rate of each region
    **Algorithm** ThresMaxDivDP - DP algorithm that returns the thresholds maximizing the divergence between key and non-key distribution.
    **Algorithm** CalcDensity - returns the region density given thresholds of the regions
    **Algorithm** OptimalFPR - returns the optimal false positive rate of the regions given thresholds
    **Algorithm** SpaceUsed - returns space used by the back-up Bloom filters given threholds and false positive rate per region.

1: **procedure** SOLVE($G_{dis}, H_{dis}, F, k$)
2:     $t \leftarrow$ ThresMaxDivDP($G_{dis}, H_{dis}, k$)               ▷ Getting the optimal thresholds for the relaxed problem
3:     $G', H' \leftarrow$ CalcDensity($G_{dis}, H_{dis}, t$)
4:     $f =$ OptimalFPR($G', H', F, k$)       ▷ Obtaining optimal false positive rates of the general problem for given thresholds
5:
6:     **return** $t$ , $f$ Array

---

# B   OPTIMAL FALSE POSITIVE RATE FOR GIVEN THRESHOLDS

We provide the pseudocode for the algorithm to find the optimal false positive rates if threshold values are provided. The corresponding optimization problem is given in Eq.18. As the boundaries for the regions are already defined, one only needs to find the optimal false positive rate for the backup Bloom filter of each region.

$$
\begin{aligned}
\min_{f_i = 1 \dots k} \quad & \sum_{i=1}^{k} \left( G(t_i) - G(t_{i-1}) \right) \times c \log_2\left(\frac{1}{f_i}\right) \\
\text{constraints} \quad & \sum_{i=1}^{k} \left( H(t_i) - H(t_{i-1}) \right) \times f_i = F \\
& f_i \leq 1 \quad i = 1 \dots k
\end{aligned}
\tag{18}
$$

Alg.1 gives the pseudocode. We first assign false positive rates based on the relaxed problem but may find that $f_i \geq 1$ for some regions. For such regions, we can set $f_i = 1$, re-solve the relaxed problem with these additional constraints (that is, excluding these regions), and use the result as a solution for the general problem. Some regions might again have a false positive rate above one, so we can repeat the process. The algorithm stops when there is no new region with false positive rate greater than one. This algorithm finds the optimal false positive rates for the regions when the thresholds are fixed.

---

**Algorithm 3** Solving the general problem

---

**Input** $G_{dis}$ - the array containing discretized key density of each region
**Input** $H_{dis}$ - the array containing discretized key density of each region
**Input** $F$ - target overall false positive rate
**Input** $k$ - number of regions
**Output** $t$ - the array of threshold boundaries of each region
**Output** $f$ - the array of false positive rate of each region
**Algorithm** ThresMaxDivDP - DP algorithm that returns the thresholds maximizing the divergence between key and non-key distribution.
**Algorithm** CalcDensity - returns the region density given thresholds of the regions
**Algorithm** OptimalFPR - returns the optimal false positive rate of the regions given thresholds

1: **procedure** SOLVE($G_{dis}, H_{dis}, F, k$)
2:    $MinSpaceUsed \leftarrow \infty$                                  ▷ Stores minimum space used uptil now
3:    $index \leftarrow -1$                        ▷ Stores index corresponding to minimum space used
4:    $G_{last} \leftarrow 0$                                ▷ Key density of the current last region
5:    $H_{last} \leftarrow 0$                      ▷ Non-key density of the current last region
6:
7:    **for** $i$ **in** $k-1, k, ... N-1$ **do**            ▷ Iterate over possibilities of last region
8:       $G_{last} \leftarrow \sum_{j=i}^{N} G_{dis}[j]$           ▷ Calculate the key density of last region
9:       $H_{last} \leftarrow \sum_{j=i}^{N} H_{dis}[j]$
10:      $t \leftarrow$ ThresMaxDivDp($G[1..(i-1)], H[1..(i-1)], k-1$)    ▷ Find the optimal thresholds for the rest of the array
11:      $t.append(i)$
12:      $G', H' \leftarrow$ CalcDensity($G_{dis}, H_{dis}, t$)
13:      $f =$ OptimalFPR($G', H', F, k$)          ▷ Find optimal false positive rates for the current configuration
14:      **if** ($MinSpaceUsed <$ SpaceUsed($G_{dis}, H_{dis}, t, f$))
15:        **then** $MinSpaceUsed \leftarrow$ SpaceUsed($G_{dis}, H_{dis}, t, f$); $index \leftarrow i$    ▷ Remember the best performance uptil now
16:
17:    $G_{last} \leftarrow \sum_{j=index}^{N} G_{dis}[j]$
18:    $H_{last} \leftarrow \sum_{j=index}^{N} H_{dis}[j]$
19:    $t \leftarrow$ ThresMaxDivDP($G[1..(index-1)], H[1..(index-1)], k-1$)
20:    $t.append(index)$
21:    $G', H' \leftarrow$ CalcDensity($G_{dis}, H_{dis}, t$)
22:    $f =$ OptimalFPR($G', H', F, k$)
23:
24:    **return** $t$ , $f$ Array

---

# C   ALGORITHM FOR FINDING THRESHOLDS

We provide the pseudocode for the algorithm to find the solution for the relaxed problem; Alg.3 finds the thresholds and false positive rates. As we have described in the main text, this algorithm provides the optimal parameter values, if $(G(t_{i-1}) - G(t_i))/(H(t_{i-1}) - H(t_i))$ is monotonically increasing.

The idea is that only the false positive rate of the rightmost region can be one. The algorithm receives discretized key and non-key densities. The algorithm first iterates over all the possibilities of the rightmost region. For each iteration, it finds the thresholds that maximize the KL divergence for the rest of the array for which a dynamic programming algorithm exists. After calculating these thresholds, it finds the optimal false positive rate for each region using Alg.1. After calculating the thresholds and false positive rates, the algorithm calculates the total space used by the back-up Bloom filters in PLBF. It then remembers the index for which the space used was minimal. The $t_i$'s and $f_i$'s corresponding to this index are then used to build the backup Bloom filters. The worst case time complexity is then $\mathcal{O}(N^3 k)$.

# D   ADDITIONAL CONSIDERATIONS

## D.1   SANDWICHING: A SPECIAL CASE

We show here that the sandwiching approach can actually be interpreted as a special case of our method. In the sandwiching approach, the learned model is sandwiched between two Bloom filters as shown in Fig.3(A). The input first goes through a Bloom filter and the negatives are discarded. The positives are passed through the learned model where based on their score $s(x)$ they are either directly accepted when $s(x) > t$ or passed through another backup Bloom filter when $s(x) \leq t$. In our setting, we note that the pre-filter in the sandwiching approach can be merged with the backup filters to yield backup filters with a modified false positive rate. Fig.3(B) shows what an equivalent design with modified false positive rates would look like. (Here equivalence means we obtain the

same false positive rate with the same bit budget; we do not consider compute time.) Thus, we see that the sandwiching approach can be viewed as a special case of the PLBF with two regions.

However, this also tells us we can make the PLBF more efficient by using sandwiching. Specifically, if we find when constructing a PLBF with $k$ regions that $f_i < 1$ for all $i$, we may assign $f_0 = \max_{1 \leq i \leq k} f_i$. We may then use an initial Bloom filter with false positive rate $f_0$, and change the target false positive rates for all other intervals to $f_i/f_0$, while keeping the same bit budget. This approach will be somewhat more efficient computationally, as we avoid computing the learned model for some fraction of non-key elements.

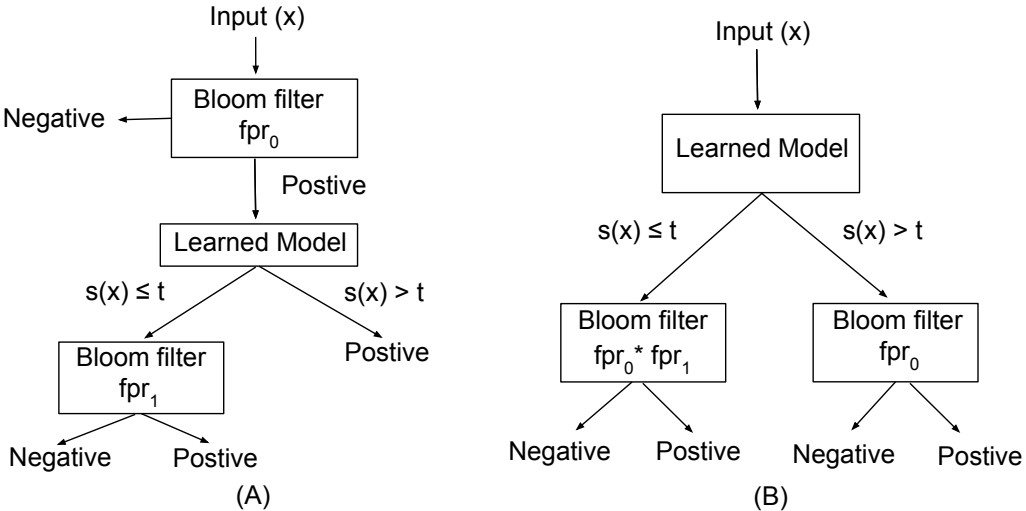

Figure 3: (A) represent LBF with sandwiching.(B) represents a PLBF design equivalent to the sandwiching approach.

## D.2 PERFORMANCE AGAINST NUMBER OF REGIONS $k$

Earlier, we saw the maximum space saved by using PLBF instead of a normal Bloom filter is linearly proportional to the $D_{KL}(g(\hat{\mathbf{t}}), h(\hat{\mathbf{t}}))$. If we split any region into two regions, the overall divergence would increase because sum of divergences of the two split regions is always more than the original divergence, as shown in Eq.19. Eq.19 is an application of Jensen's inequality.

$$\left((g_1 + g_2) \times \log \frac{(g_1+g_2)}{(h_1+h_2)}\right) \leq \left(g_1 \times \log \frac{g_1}{h_1}\right) + \left(g_2 \times \log \frac{g_2}{h_2}\right) \tag{19}$$

Increasing the number of regions therefore always improves the maximum performance. We would hope that in practice a small number of regions $k$ would suffice. This seems to be the the case in our experience; we detail one such experiment in our evaluation(4.2).

## D.3 PERFORMANCE USING VARIOUS BLOOM FILTER VARIANTS

We consider how the space saved of the PLBF varies with the type of backup Bloom filter being used. The PLBF can use any Bloom filter variant as the backup Bloom filter. When we compare our performance with a Bloom filter variant, we use that same Bloom filter variant as the backup Bloom filter for a fair comparison.

First, absolute space one can save by using a PLBF instead of a Bloom filter variant is given in Eq.10. This quantity increases with increasing $c$[6].

---

[6]The sizes of standard Bloom filter variants are proportional to $|S| \times \log_2(1/f)$, where $S$ is the set it represents, and $f$ is the false positive rate it achieves. See e.g. Mitzenmacher (2018) for related discussion. The constant $c$ depends on which type of Bloom filter is used as a backup. For example, $c = \log_2(e)$ for standard Bloom filter, $c = 1.0$ for the optimal Bloom filter.

The relative space one saves by using PLBF instead of the given Bloom filter variant is shown in Eq.20. This quantity is the ratio of the space saved by PLBF (as shown in Eq.10) divided by the space used by the given Bloom filter variant ($c \times |S| \times \log_2(1/F)$) as shown in Eq.20.

$$\frac{\left(c \times |S| \times D_{KL}\left(g(\hat{\mathbf{t}}), h(\hat{\mathbf{t}})\right) - \text{Size Of Learned Model}\right)}{c \times |S| \times \log_2(1/F)} \tag{20}$$

Cancelling the common terms we obtain the following Eq.21.

$$\left(\frac{D_{KL}\left(g(\hat{\mathbf{t}}), h(\hat{\mathbf{t}})\right)}{\log_2(1/F)} - \frac{\text{Size Of Learned Model}}{c \times |S| \times \log_2(1/F)}\right) \tag{21}$$

The relative space saved, like the absolute space saved, also increases with increasing $c$. Thus, both the relative and absolute space saved for the PLBF is higher for a standard Bloom filter ($c = 1.44$) than an optimal Bloom filter ($c = 1.00$), and hence our experiments in Section 4.1 are conservative estimates of gains possible in practice using PLBF.

## E    ADDITIONAL EXPERIMENTS

### E.1    PERFORMANCE W.R.T STANDARD BLOOM FILTERS

Earlier, we evaluated our performance using optimal Bloom filters and here we present results using standard Bloom filters. As shown in Appendix.D.3, PLBF performs better w.r.t standard Bloom filters than optimal Bloom filters. As one can see from Fig.4, PLBF performs better than the standard Bloom filter.

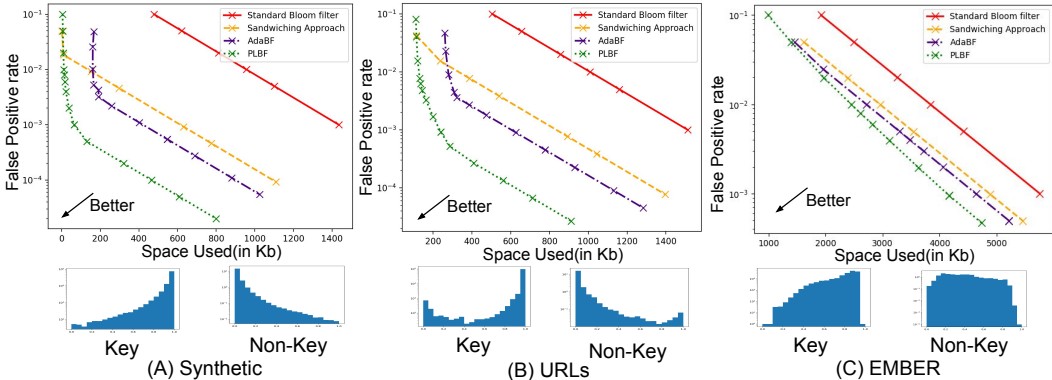

Figure 4: FPR vs Space for the (A) Synthetic (B) URLs (C) EMBER datasets for various baselines along with key and non-key score distributions.

### E.2    PERFORMANCE AND MODEL QUALITY

Here we provide an experiment to see how the performance of various methods varies with the quality of the model. As discussed earlier, a good model will have high skew of the distributions $g$ and $h$ towards extreme values. We therefore vary the skew parameter of the Zipfian distribution to simulate the model quality. We measure the quality of the model using the standard F1 score. Fig.5(B) represents the space used by various methods to achieve a fixed false positive rate of $0.001$ as we vary the F1 score of the model. The figure shows that as the model quality in terms of the F1 score increases, the space required by all the methods decreases (except for the optimal Bloom filter, which does not use a model). The space used by all the methods goes to zero as the F1 score goes to 1, as for the synthetic dataset there is no space cost for the model. The data point corresponding to F1 score equal to 0.99 was used to plot Fig.2(A).

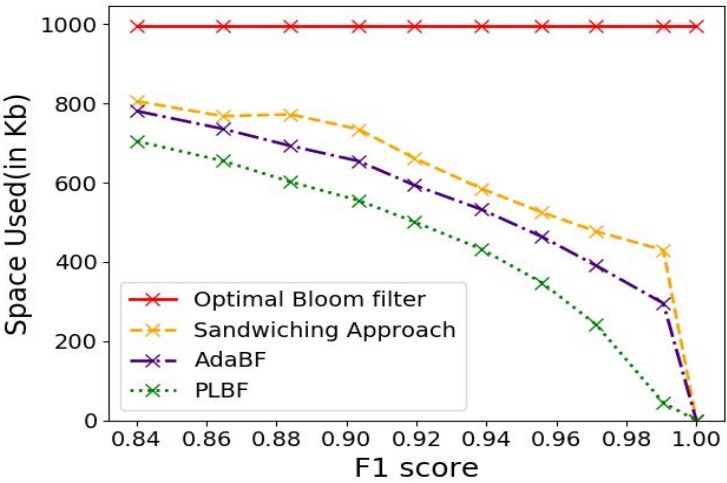

Figure 5: Space used by various baselines as we increase F1 score for Synthetic dataset

### E.3    DISCRETIZATION EFFECT ON DYNAMIC PROGRAMMING RUNTIME, PLBF SIZE

All the runtime experiments in this subsection and the next subsection are measured using an 2.8GHz quad-core Intel Core i7 CPU @ 2.80GHz with 16GB of memory. We use the *bloom-filter* python package [bloom filter] for our backup Bloom filters. The dynamic programming algorithms are implemented in Python.

Here we provide an experiment to see how the dynamic programming (DP) algorithm runtime (psuedo code in Alg.3) and PLBF size vary with level of discretization ($N$). In the tables below, we have the DP algorithm runtime and space taken by the PLBF to achieve an approximate empirical false positive rate of $0.001$ for various $N$. As discussed in Sec. 3.3.2, with increasing value of $N$ one gets closer to optimal parameters, at the cost of higher computation time. This trend is demonstrated in the table below for the URLs and EMBER datasets. We note that if runtime is an issue, the increase in size from using smaller $N$ is relatively small.

| N | DP Runtime(in sec) | PLBF Size (in Kb) |
|---|---|---|
| 50 | 1.17 | 187.6 |
| 100 | 2.15 | 184.37 |
| 500 | 10.97 | 183.63 |
| 1000 | 26.94 | 183.55 |
| 2000 | 56.79 | 182.85 |

Table 1: DP runtime and space used by PLBF as we increase the discretization $N$ in the URLs dataset

| N | DP Runtime(in sec) | PLBF Size (in Kb) |
|---|---|---|
| 50 | 1.36 | 2952.33 |
| 100 | 2.52 | 2944.68 |
| 500 | 11.39 | 2933.09 |
| 1000 | 25.26 | 2928.76 |
| 2000 | 56.12 | 2926.79 |

Table 2: DP runtime and space used by PLBF as we increase the discretization $N$ in the EMBER dataset

### E.4    CONSTRUCTION TIME FOR VARIOUS BASELINES

Here we look at the construction time breakdown for the PLBF and various alternatives, with the goal of seeing the cost of in terms of construction time for using the more highly tuned PLBF. The construction time of all the learned Bloom filters includes the model training time and parameter estimation time, which are not required for the standard Bloom filter construction process. Since

we use the same model for all learned baselines, the model construction time is the same for all of them. In Fig.6, we plot the construction time breakdown for various baselines in order to achieve an approximate empirical false positive rate of $0.001$. Recall that the AdaBF and Sandwiching approaches use heuristics to estimate their parameters and unsurprisingly they therefore seems somewhat faster than PLBF. However, for $N = 100$ we see the parameter estimation time is smaller than the key insertion time and model training time. The parameter estimation time for PLBF varies with the level of discretization we use for the DP algorithm. The PLBF with $N = 1000$ takes the longest to execute while standard Bloom filter is fastest baseline. As shown in Table1 above, using $N = 1000$ gives only a slight improvement in size. We therefore believe that if construction time is an issue, as for situations where one might want to re-learn and change the filter as data changes, one can choose parameters for PLBF construction that would still yield significant benefits over previous approaches.

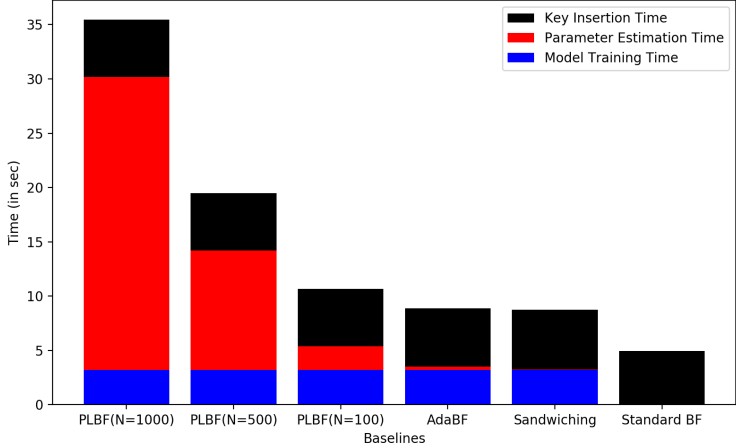

Figure 6: Construction time breakdown for various baselines for the URLs dataset

