# OpenReview forum: "Partitioned Learned Bloom Filters"
_ICLR.cc/2021/Conference — ICLR 2021 Poster_

### Official Review · AnonReviewer4 · 2020-10-27
**Fine tuning partitioned learned Bloom filter**

**Rating:** 6
**Confidence:** 3

**Review:**


This work proposed a technique to fine tune the partitioned learned Bloom filter to reduce the space consumption given a false positive rate threshold.

The idea is to formulate the problem into a two-part optimization problem: How to best partition the scores from the model into a given number of regions and how to choose thresholds for the regions to minimize the overall space consumption of Bloom filters. A relaxed version of the latter problem is addressed by using KKT conditions to obtain the optimal thresholds. The former problem is addressed by discretizing the region boundaries and dynamic programming.

Overall, I like the idea of fine tuning the partitioned learned Bloom filter. It seems to me that it will also be possible to fine tune the number of partitions as well, e.g., a simple way is to use a binary search. The evaluation result is impressive compared with baselines in terms of space consumption vs. false positive rate. And the writing of the paper is clear and easy to follow.

Having said that, I have some concerns with this line of work.

First, IMHO, the real challenge of putting these learned Bloom filters into work is how to maintain them under insertions. While it is OK that this is not the focus of this work, it seems to me that the proposed optimal learned Bloom filter can be brittle under insertions. It will be great to understand how the proposed technique degrades compared with the baselines. In addition, with partitioned Bloom filters, it seems to be more prone to resizing upon insertions compared with using a single Bloom filter.

Second, it is possible that the proposed technique needs to be reoptimized upon insertions. In this case, it will be important to understand the overhead of constructing the Bloom filters proposed in this technique compared with the baselines. However, the overhead of constructing the various variants of Bloom filters is missing in the evaluation. It might be possible to reduce the overhead by using a coarser grained discretization for the DP. The performance, however, can degrade with a coarser grained discretization.

---

> ### Author Response · Authors · 2020-11-17
> **Response to Reviewer 4**
>
> We thank the reviewer for their comments.
>
> 1) We first note that, if insertions are an issue, they sometimes can be dealt with without significant changes.  For example, if the number of insertions are small, one can insert new set items by adding to the corresponding Bloom filter while only slightly increasing the false positive rate.  Alternatively, there is another paper which deals with the issue of re-training models during insertions that provides additional insight: Adaptive Learned Bloom Filters under Incremental Workloads. If we keep the optimal parameters fixed, the backup Bloom filters may need to be resized as keys are inserted in them. When inserting keys, though the number of resizes of the backup Bloom filters would be more but the time would be similar to resizing a standard Bloom filter.  The reviewer may be correct that one might need to resize Bloom filters slightly more often than a single Bloom filter, because of variability in where insertions occur.  (Do note though that the insertions may be spread among the multiple backup Bloom filters, so it need not be dramatically different.)  However, if one resizes individual Bloom filters, then the time to resize will be smaller, as each individual Bloom filter is smaller than a single large Bloom filter.
>
> 2) The construction time of PLBF is more than the standard Bloom filter owing to the extra steps of model construction and the optimal parameter computation. The model construction time is a common step for all the learned techniques but the parameter computation time varies across them.
>  We have added experiments in Appendix E 3-4, that show the initial Bloom filter construction time for various baselines and the effect of the level of discretization(N) on the optimisation overhead and PLBF size.

---

### Official Review · AnonReviewer2 · 2020-10-28

**Rating:** 7
**Confidence:** 3

**Review:**

The paper proposes a generalization of the sandwiched Bloom filter model that maintais a set of score partitions instead of just two and an algorithm for optimizing parameters of the partition under the target false-positive rate. Authors evaluate partitioned Bloom filter on three datasets and demonstrate that delivers better false positive rates under for a given model size compared to the baselines.

I find the paper quite innovative and the experimental results impressive. My main concern is regarding the paper clarity.
1. How can a region's FPR $f_i$ be greater than 1? Perhaps, this is something very obvious, but I couldn't get this immediately and perhaps some other readers might  struggle here too.
2. The learned model size appears in the optimization objective, but is considered given. I wonder what will change if we allow to trade-off the learned model power for a larger number of regions and larger backup Bloom filters in each region? Does it even make sense to ask such a question? Would the results change if a different model is used? I think it is possible that for a given variant of a learned Bloom filter, a different model may result into different values of optimal parameters and a difference performance, thus these should ideally be optimized independently for each of the baselines. I also think that size of the pickle file is arguably not the best estimate for the learned model size if indeed a different model is used for each of the filters, e.g. a neural network might admit a decent compression rate if a lower precision number format is used etc. Thus, it is important to separate the impact made by a learned model from an algorithmic improvement.
3. Is there any variance caused by observing particular distributions G and H? Is it small enough to ignore or confidence interals for each of the curves might actually overlap? I would also be interested in understanding behaviour of all considered models as the sample size changes.

I also feel like authors can cite *Meta-learning neural Bloom filters. Rae et al, 2019* as it considers a relevant (although a different as well) setting.

Nevertheless, I think the kind of analysis presented in the paper very useful for the community and for further development of learned data structures. I recommend acceptance and I will gladly raise my rather conservative score if authors could clarify the points mentioned above.

---

> ### Author Response · Authors · 2020-11-17
> **Response to Reviewer 2**
>
> We would like to thank the reviewer for their insightful comments.  In particular, we are confident that we can address all the issues raised.
>
> 1) As noted by the reviewer, the false positive rate cannot be greater than 1 for any region, and this constraint is added in our optimization problem as shown in Eq(4). In order to solve the optimization problem, we relax the constraint to allow that the fpr variables can be larger than 1. This might lead to a solution to the mathematical optimization problem where fpr >1 for some regions, which then needs to be corrected, by setting the fpr for that region to 1 and re-optimizing, as we explain in Section 3.3.3.
>
> 2) We solve the problem assuming we have been provided a model already trained on the data. As you correctly pointed out, there exists a trade-off between the size of the model and the algorithmic improvement it provides. The best performance of the learned model is proportional to KL divergence between G and H. With increasing model size, this quantity improves leading to smaller backup Bloom filters, which can create a trade off between the model size and the backup Bloom filter size.  As noted in another review comment, if we had some meta-model that could relate the model size and the model accuracy, we could include that into our optimization design, but we are unaware of such meta-models.
> We used the uncompressed pickle file size for the experiments as it represents the size of the random forest model accurately, which is what we used in our experiments. We agree for other models one would use a size metric that represented that model.
>
> 3) Since the key set is fixed, the number of keys in each region remains the same. We estimate the H distribution using a sample of non-keys leading to variation between real and estimated H. The estimation is robust for larger sample sizes.  (This is discussed in previous work, e.g., in Theorem 4 of [A Model for Learned Bloom Filters, and Optimizing by Sandwiching])
>
> We have added Meta-learning neural Bloom filters. Rae et al, 2019 as a related work.

---

### Official Review · AnonReviewer1 · 2020-11-02
**A somewhat descriptive scheme for constructing memory-efficient partitioned bloom filters incorporating a learned model.**

**Rating:** 7
**Confidence:** 3

**Review:**

A clear exposition of the problem and proposed solution, the paper key strength is in the formulation of the partitioned bloom filter as an optimization problem that generalizes previously proposed architectures, and prescribes an interpretable solution for the choice of the optimal partition-thresholds in terms of the properties of the given learned model (specifically, its false-negative and true-negative threshold dependent curves). Furthermore, the experimental section clearly illustrates a significant advantage of the proposed method over state of the art alternatives.
The theoretical treatment, however may be considered as a first attempt at combining essential properties of a learned model and bloom-filter design - the paper could be improved by considering and motivating the usage of a particular lower bound for the space cost of a bloom filter and incorporating in the the optimization formulation the relation between the size of the learned model and the qualities of the false positive and false negative curves. The latter, especially, makes a significant practical difference since the model may be part of the design and assuming out the size-quality tradeoff results in a sub-optimal scheme and renders the overall proposed solution as a heuristic still.

Update (Nov 30th) In light of the author's responses and the other reviews I increase my score for this paper to 7: Good paper, accept.

---

> ### Author Response · Authors · 2020-11-17
> **Response to Reviewer 1**
>
> We thank the reviewer for the comments.  The reviewer suggests that the current analysis takes the model as a given where it could be part of the design.  This would be the case if we had some meta-model that could relate the model size and the model accuracy;  if we had such a meta-model, we could include that into our optimization design.  However, we are not aware of any suitable model for model size vs. accuracy in these domains.  We therefore agree with the reviewer that this would be an interesting direction for future work.  [However, we expect in most situations this will be a lower order effect, in that the model must be sufficiently large to be reasonably accurate, but will at some point have diminishing returns.]  Regardless of whether this can be incorporated into the design phase, we believe the optimization framework we have presented shows clear payoffs over previous works, and in particular shows how to obtain near-optimal performance for any given model.

---

### Decision · Program_Chairs · 2021-01-07
**Final Decision**

**Decision:**

Accept (Poster)

**Comment:**

All of the reviewers thought that this paper addresses an interesting and important problem.  Several of the reviewers thought that the paper gave a creative approach for training bloom filters and this would be of interest to the community.